# Copper (I) or (II) Replacement of the Structural Zinc Ion in the Prokaryotic Zinc Finger Ros Does Not Result in a Functional Domain

**DOI:** 10.3390/ijms231911010

**Published:** 2022-09-20

**Authors:** Martina Dragone, Rinaldo Grazioso, Gianluca D’Abrosca, Ilaria Baglivo, Rosa Iacovino, Sabrina Esposito, Antonella Paladino, Paolo V. Pedone, Luigi Russo, Roberto Fattorusso, Gaetano Malgieri, Carla Isernia

**Affiliations:** 1Department of Environmental, Biological and Pharmaceutical Science and Technology, University of Campania “Luigi Vanvitelli”, Via Vivaldi 43, 81100 Caserta, Italy; 2Institute of Biostructures and Bioimaging, National Research Council (IBB-CNR), Via Pietro Castellino 111, 80131 Naples, Italy

**Keywords:** copper, metal-binding proteins, zinc finger, protein misfolding, binding affinity, metal ion toxicity

## Abstract

A strict interplay is known to involve copper and zinc in many cellular processes. For this reason, the results of copper’s interaction with zinc binding proteins are of great interest. For instance, copper interferences with the DNA-binding activity of zinc finger proteins are associated with the development of a variety of diseases. The biological impact of copper depends on the chemical properties of its two common oxidation states (Cu(I) and Cu(II)). In this framework, following the attention addressed to unveil the effect of metal ion replacement in zinc fingers and in zinc-containing proteins, we explore the effects of the Zn(II) to Cu(I) or Cu(II) replacement in the prokaryotic zinc finger domain. The prokaryotic zinc finger protein Ros, involved in the horizontal transfer of genes from *A. tumefaciens* to a host plant infected by it, belongs to a family of proteins, namely Ros/MucR, whose members have been recognized in different bacteria symbionts and pathogens of mammals and plants. Interestingly, the amino acids of the coordination sphere are poorly conserved in most of these proteins, although their sequence identity can be very high. In fact, some members of this family of proteins do not bind zinc or any other metal, but assume a 3D structure similar to that of Ros with the residues replacing the zinc ligands, forming a network of hydrogen bonds and hydrophobic interactions that surrogates the Zn-coordinating role. These peculiar features of the Ros ZF domain prompted us to study the metal ion replacement with ions that have different electronic configuration and ionic radius. The protein was intensely studied as a perfectly suited model of a metal-binding protein to study the effects of the metal ion replacement; it appeared to tolerate the Zn to Cd substitution, but not the replacement of the wildtype metal by Ni(II), Pb(II) and Hg(II). The structural characterization reported here gives a high-resolution description of the interaction of copper with Ros, demonstrating that copper, in both oxidation states, binds the protein, but the replacement does not give rise to a functional domain.

## 1. Introduction

The mechanism through which the right metal is integrated into the correct protein, i.e., the metal-binding selectivity of proteins, still needs investigation. The affinities for the metal binding of natural proteins should follow the Irving–Williams series. However, metal-binding proteins are capable of overcoming the Irving–Williams series restrictions binding metals other than copper.

Metal-binding selectivity can be controlled by dedicated chaperones delivering the metal to the correct cellular sites or by the availability of the right metal at the location of protein folding [1,2]. In proteins with metal-binding sites suitable for different types of ions, the distribution of metal ions in the cellular compartment in which the protein folds appears to have great influence on metal specificity [3].

The situation inside cells is also complicated (especially with respect to the situation in a dilute buffer system) by the highly crowded environment that increases nonspecific interactions and by the fact that proteins follow different routes (e.g., many proteins need to be transported through membranes often in their unfolded state) depending on their final target compartment [4]. Thus, to pinpoint the interplay between protein folding and the right metal acquisition, the spatial and temporal location of the protein as well as the concentration of the metal ion in the specific cellular compartment should be considered [5]. 

A strict interplay is known, for instance, to involve copper and zinc. Involved in key processes, copper is an essential trace element whose uncontrolled reactivity as well as its ability to bind at sites for other metals makes it toxic. Therefore, living systems have developed fine copper-transport systems to facilitate the specific delivery of copper to target proteins [6]. Theoretically, zinc is one of copper competing metals and is necessary to many more proteins than copper [7,8]. Zinc cellular levels are similarly thoroughly controlled and the mechanism of its delivery to targets is not well understood (i.e., no cytoplasmic metallo-chaperones are known so far) [9,10,11].

Both metal ions have arisen as non-structural intracellular mediators of cell signaling. This role, particularly in the case of copper (both as Cu^+^ and Cu^2+^), is quite interesting as it was previously thought to be specific of redox-inactive metals such as K^+^, Na^+^ and Ca^2+^ and, more recently, also Zn^2+^.

In vitro, the roles of copper ions in protein folding and stability can be radically different: copper stabilizes azurin, while its binding to beta-2-microglobulin strongly destabilizes the protein [12]. In some cases, even if the structure is stabilized by copper binding, the protein becomes nonfunctional, as in the zinc finger protein Sp1. Cu(I) possesses a strong binding affinity to Sp1 that renders it capable to substitute the zinc ion in this zinc finger. Such substitution causes only small structural alterations and forms a well-folded protein; however, it is unable to bind its cognate DNA [13]. It is also known that copper is able to bind specifically unfolded and partially-folded structures in vitro, thereby resulting in folding reactions becoming “toxic” [14,15,16].

Toxicological exposure to copper or a breakdown in the homeostatic mechanism that regulates the amount of this metal present in a cell can result in abnormal cellular mechanisms [17]. In these conditions, proteins interact with concentrations of copper different from the physiological conditions often leading to protein inactivation, misfolding and aggregation [18]. Misfolding and aggregation are at the basis of many diseases and there is a plethora of data indicating that abnormal concentrations of metal ions are capable of accelerating these processes [19,20].

The selectivity of a copper-binding site in a protein is only partially dictated by the chemical identity, number and geometry of the ligating groups, as protein architectures can finely tune coordination sites to gain precise selectivity and thermodynamic affinities [21]. These factors define the relative stabilities of the resultant Cu(I) and Cu(II) complexes. Exposure to “toxic” copper could, in theory, alter the function of any metallo-protein [22].

In this context, we propose the study of the influence that the substitution of the native zinc ion by Cu(II) or Cu(I) has on the structure and folding of the prokaryotic zinc finger protein Ros87, the DNA binding domain of the *A. tumefaciens* protein Ros [23,24,25].

Ros87 is a small (87 residues) protein that belongs to the Ros/MucR family [25,26,27,28]. It encompasses two α-helices and three antiparallel β-strands folded into a 60-residue globular structure that buries a hydrophobic core made by the side chains of 15 residues; such a compact domain is surrounded by two disordered flanking regions [29] (Figure 1).

The presence of a structural zinc ion coordinated by the side chains of two cysteines and two histidines is crucial for the structural stability of this protein and is determinant in its folding process [30,31,32]. As the histidine imidazole and the cysteine thiolate belong to the array of amino acid functional groups more commonly found to bind copper ions in the cell [21], Ros87 represents an excellent model for our biophysical binding/folding investigation [33,34,35,36,37,38]. Our work aimed at contributing to the study of factors that convey the metal-binding selectivity in proteins and the deleterious effects of toxic metals on metallo-proteins [39].

## 2. Results

### 2.1. Cu(I) and Cu(II)-Ros87 Binding Affinity

The study of copper (I) and (II) binding to Ros87 started with the evaluation of the magnitudes of these interactions. Both metals give a ligand-to-metal charge-transfer (LMCT) band in the UV-Vis spectra that is diagnostic of the binding. For this reason, we decided to perform UV-Vis spectrophotometric titrations to obtain the Kds for both metal ions. To avoid overestimations, we carefully optimized the protein concentrations, thus obtaining a binding isotherm in both cases.

Titration with Cu(I) was carried out in Tris buffer at pH = 6.5, in the presence of TCEP to avoid cysteine oxidation, but also to keep the copper reduced to Cu(I) [41]. The success in reducing the copper was confirmed by the presence of the characteristic UV-Vis bands at 263 and 300 nm reported in the literature [42] that also confirmed that the oxidation state, carefully monitored in time, remained unchanged throughout the time of the experiment. In the case of Cu(II), titration was performed in the same buffer without TCEP. To avoid cysteine oxidation, the sample preparation was done in the presence of TCEP. The final step of the preparation foresaw a fast dialysis versus the TCEP-free buffer. Only freshly prepared samples were used and a titration with Co(II) [43] of aliquots of the same samples used for Cu(II) titration confirmed the absence of oxidized cysteines.

The metal ions were added to a solution of Ros87 till the plateau of the titration curve was clearly reached (that is Cu(I)/Ros87 1.8/1 and Cu(II)/Ros87 2.2/1 molar ratio). The spectra were recorded directly upon each addition but after a time (usually 5 min) that assured that the reaction equilibrium was reached.

In Figure 2A,B, the UV-Vis spectra and the plot of absorbance versus concentration of Cu(I) are reported. The LMCT Cu(I)-S band at 263 nm [42] was followed and clearly indicates the formation of a complex with a 1:1 stoichiometry. Using the 1:1 model to fit the UV data, we obtained an apparent Kd (Kd*) of 6.8 (±2.5) × 10^−8^ M. It should be underlined here that this value is an apparent Kd (Kd*) because of competition with the Tris buffer [44]. In fact, this buffer, which is very commonly used in the study of proteins, is known to have some affinity for the metal ions studied in this article [45]. Interestingly, the extinction molar coefficient for Cu(I)-Ros87 gives a value of ca 6000 m^2^ mol^−1^, suggesting that only one cysteine thiol is tightly bound to the metal involved in the coordination [42].

Furthermore, the Cu(I)-Ros87 (1.8:1 ratio) solution was retro-titrated with a solution of Zn(II) (Appendix A) and the disappearance of the band at 263 was followed. The apparent dissociation constant for Zn(II)-Ros87 (4.2 (±0.45) × 10^−10^ M) was calculated and gave values similar to those already published [32]. The direct titration of Apo-Ros87 with Cu(II) is reported in Figure 2C together with the plot of absorbance versus the concentration of Cu(II) (panel D). In this case, the band at 236 nm, typical of a coordination of this metal ion to the histidine imidazole (LMCT Cu(II)-Im [46], was followed that clearly indicates the formation of a complex with a 1:1 stoichiometry and gave a calculated Kd* = 4.5 (±0.9) × 10^−7^ M. Also in this case, Cu(II)-Ros87 (1.8:1) solution was retro-titrated with a solution of Zn^2+^ (Appendix A) and the disappearance of the band at 236 was followed. The apparent dissociation constant for Zn(II)-Ros87 (8.3 (±0.96) × 10^−10^ M) was calculated and gave values similar to those already published (5.8 × 10^−10^).

Copper, in both oxidation states, is capable of binding Ros87 with a good affinity that is at least two orders of magnitude lower compared to the affinity for its wildtype zinc ion. This allows the exclusion of the possibility for copper ions to replace the structural zinc ion within the structure of this zinc finger protein.

### 2.2. Secondary Structure Evaluation of Cu(I) and Cu(II)-Ros87

To evaluate the effect of the copper binding on Ros87 structure, we performed a CD analysis of the protein in the presence of Cu(I) or Cu(II) in a 1:1 stoichiometry ratio. In both cases, the normalized data were compared with the same spectra obtained for Apo-Ros87 (i.e., Ros87 in which the Zn^2+^ has been removed) and with the 1:1 Zn(II)-Ros87 complex (Figure 3). As evidenced in Figure 3, Zn(II)-Ros87 gives rise to a CD spectrum typical of a well-structured protein with a significant secondary structure content. In contrast, the spectra of Cu(I)-Ros87 and Cu(II)-Ros87 suggests a much lower amount of secondary structure elements. Interestingly, the two metal ions have a different effect on the protein structure: the Ros87 folding degree depends on the copper oxidation state. Overall, the data indicate that Cu(I)-Ros87 and Cu(II)-Ros87 conformational behaviors fall between the two extremes constituted by the structured Zn(II)-Ros87 on one side and unstructured Apo-Ros87 on the other. Interestingly, although both metals give substantial perturbation in the structure, in the case of the Cu(II) complexation, a minor stabilizing effect in terms of the protein structure is visible with respect to Cu(I), as the protein in this latter case appears to have a slightly higher secondary structure content.

It is also interesting to underline how, in the case of Cu(I)-Ros87, the protein obtained with the back-titration with the native Zn(II) gives back (Appendix A) a CD spectrum mostly identical to the wildtype Zn(II)-Ros87, while in the case of the back-titration of Cu(II)-Ros87, the spectrum of the obtained zinc protein is different. This different behavior suggests, for this latter case, the probable formation of intramolecular disulfide bonds involving the cysteine residues.

### 2.3. NMR Structural Characterization of Cu(I) and Cu(II)-Ros87

Figure 4A,B reports the NMR characterization of Cu(I)-Ros87: the superposition of the proton–nitrogen HSQC spectrum for this protein with equivalent proton-nitrogen HSQC spectra recorded for the apo-protein and Zn(II) bound Ros87 under identical conditions. In marked contrast to the spectrum of Zn(II)-Ros87, the resonances in the spectrum of the Cu(I)-bound protein are poorly dispersed, covering a range of less than 1 ppm in the ^1^H dimension and less than 25 ppm in the ^15^N dimension. This absence of dispersion is not consistent with a well-folded structure stabilized by extensive tertiary interactions, indicating that in Cu(I)-Ros87, many of the amino acids are unprotected from the solvent and consequently experience a non-unique local environment. Despite the low degree of dispersion, many resonances in the spectrum are well resolved due to the narrow line widths and sharpness of the signals caused by a high conformational mobility within the protein, indicative of largely unfolded proteins or peptides [47]. Nonetheless, despite the paucity of the secondary structure, both the CD and NMR data nicely reconcile in demonstrating that upon copper binding, minor structural rearrangements of the protein can be observed; the NMR spectra for the Cu(I)-substituted protein clearly show important chemical shift differences in the proton chemical shift with respect to the apo-protein. However, these rearrangements are not sufficient to obtain a fully structured protein. The ^1^H-^15^N-HSQC spectrum for Cu(II)-Ros87 is reported in Appendix A. As expected, due to the Cu(II) paramagnetic relaxation effect, the spectrum of Cu(II)-Ros87 contains a smaller number of resonances compared to the Cu(I)-Ros87 spectrum. However, the distinguishable signals are almost superimposable to those of Cu(I)-Ros87 and therefore the dispersion of the signals is also comparable to that of Apo-Ros87 in this case.

We then evaluated the correlation between ^1^H_N_ and ^15^N shifts for both copper-complexed proteins and Apo-Ros87 by applying the following strategy: the ^15^N chemical shifts were weighted by a scaling factor α (αApo-Ros87 = 0.03, αCu(I)-Ros87 = 0.042, αCu(II)-Ros87 = 0.037 and Zn(II)-Ros87 = 0.10) (Figure 5A). Overall, as suggested by the scaling factors and illustrated by the correlation plot (Figure 5A), the addition of both metals induces similar (also to the apo-protein), but not identical, signals dispersion indicating that the protein in the presence of both metals does not adopt a completely unfolded conformation. Both metal ions induce slight, but different, local structural rearrangements, which presumably involve the regions surrounding the metal-binding site. Thus, to gain information on the metal-binding site, we evaluated the involvement of the histidine side chains in the metal ion coordination by analyzing both their tautomeric and protonation states acquiring a ^1^H-^15^N-HSQC experiment, in which only H_N_/N two-bond correlations are reported. This spectrum, reported in Figure 6 superimposed to the same spectrum for Zn(II)-Ros87, shows that the side chains of the histidine residues in Cu(I)-Ros87 do not experience tautomerism, while on the contrary, they are blocked in a single tautomeric form. This behavior, although not conclusive with respect to the participation of the histidines in the metal ion coordination, is compatible with such involvement.

The same experiment recorded in the presence of the Cu(II) ion returned an empty spectrum in accordance with the involvement of the histidine residues in Cu(II) binding.

However, NMR is capable of providing additional detailed information about the structural properties of even largely unfolded states. Thus, in order to evaluate the possibility of a metal-mediated protein aggregation, we resorted to estimating the relaxation rates R2 by analyzing the ^15^N linewidth in the [^1^H-^15^N] HSQC spectrum. The obtained values are reported in Figure 5B compared to the same values for Apo- and Zn(II)-Ros87. As the R2 reports on the dynamic proprieties of the proteins in solution, the fact that the behavior of both metal-bound complexes again falls between the extremes constituted by the structured monomeric Zn(II)-Ros87 and unstructured monomeric Apo-Ros87 rules out a metal-mediated aggregation of the protein.

Although the structural data indicated the lack of the formation of an ordered tertiary structure in presence of both metals, we decided to test the ability to bind the DNA, at least of Cu(I)-Ros87 (Figure 7). In order to investigate the ability of Cu(I)-Ros87 to bind the DNA, the protein Cu(I)-Ros87 was tested by EMSA using the 40 bp Ros box as a target site. The results show that Cu(I)-Ros87 cannot bind the DNA under any of the conditions tested.

## 3. Discussions

Metal binding and protein folding are highly coupled events toward metallo-protein stabilization in biologically active structures. Nature exploits the advantageous free-energy impact of metal-ion coordination to shape the protein’s proper structure, obtaining the metal-induced folding event. The rules behind the choice of the right metal driving the correct protein fold still need to be investigated. In fact, natural proteins are capable of overcoming the Irving–Williams series restriction to select the appropriate metal ion. The identity and the number of metal-interacting residues are not sufficient to explain the selection by a protein of one metal ion rather than another.

Many metals are known to have fundamental roles in organisms, both as a free ion or coupled to proteins. The importance of copper and zinc is strongly related to metallo-proteins. In this respect, copper plays two important roles: structural stabilization and redox activity. The latter is utilized by enzymes that catalyze electron transfer reactions in a broad range of metabolic activities including, for instance, energy production and antioxidant defense. Zinc is the structural and/or catalytic core of countless proteins implicated, among others, in transcriptional regulation, reactive oxygen species detoxification and carbohydrate oxidation [48]. Although these metals are vital for the correct functioning of cells, their availability is tightly controlled as their excess has been proven to be toxic [49]. Homeostatic mechanisms preserve steady levels of the essential metals and remove the toxic ones from the cell. Toxic copper is able to interact tightly with cysteines containing protein including zinc finger domains [50]. This interaction has a strong functional and structural impact on the protein. Thus, copper toxicity, such as the results of its interaction with zinc-binding proteins, is of great interest, as interferences with the DNA-binding activity of zinc finger proteins are associated with the development of a range of diseases [51]. The biological impact of copper depends on the chemical properties of its two common oxidation states: Cu(I) with a d10 electronic configuration and Cu(II) with a d9.

The first is the main form of copper under the reducing intracellular condition. Cu(I) usually interacts with proteins through histidine, cysteine or methionine residues in accordance with the HSAB (Hard and Soft Acids and Bases) classification: Cu(I) is a soft acid, a large ion with a low charge, whose binding is favored by soft or borderline ligands such as sulfur and nitrogen donor atoms found in these amino acids; conversely, Cu(II) is more of a border line acid, and as such, it also prefers the same amino acid side chains. The differences between the two ions, according to Ligand Field Theory, can be found in their preferred geometries of coordination based on the numbers of valence d electrons. Thus, the impact of the two metal ions on zinc finger and zinc-binding proteins in general is expected to be different. The interaction of Cu(I) on zinc fingers has been analyzed in detail using CP-1-based peptides with different coordination spheres: Cys2His2, Cys3His and Cys4. In all cases, the Cu(I) ion was capable of displacing the native Zn(II) ion, resulting in unstructured and likely nonfunctional domains, with the number of Cu(I) ions bound to the peptide increasing with the number of cysteines [42]. On the other hand, Yuan and co-workers demonstrated the capability of Cu(I) to displace Zn(II) in Sp1, giving a well-folded copper finger protein unable to bind its target DNA [13]. Another study has shown that Zn(II) displacement by Cu(I) did not affect the structure and DNA-binding affinity of the non-classical zinc finger domain of transcription factor CRR1 (the copper response regulator 1) [52].

The interaction of Cu(II) ion with zinc fingers from the estrogen receptor and XPA DNA repair proteins has demonstrated the ability of this ion to alter the structure and function of the zinc finger domain [53]. Thus, the possibility of zinc-to-copper substitution occurring differs among the different zinc fingers along with other zinc metallo-proteins. Moreover, the binding behavior of synthetic zinc finger peptides may not directly reproduce the binding properties of larger protein constructs. Thus, substantial differences can be found between the reported studies and our model system, with the naturally occurring Ros87 being a larger domain.

In fact, in our study on Ros87 reported here, both Cu(I) and (II) metal ions do not give a folded and/or functional protein. In the case of Cu(I), the UV-Vis data, integrated with high-resolution NMR analysis, clearly indicate the formation of a monomeric complex with a Cu(I)/protein stoichiometry of 1:1. The UV-Vis data are also clear in indicating the involvement of only one cysteine residue in the complexation of the metal ion, while the data from the ^1^H-^15^N-HSQC-type experiments, reporting H_N_/N two-bond correlations, are compatible with the involvement of the histidine residues. In particular, Ros87 shows a lower binding affinity for Cu(I) with respect to that measured for the native zinc ion in the same conditions. As a result, the titration of Cu(I)-Ros87 with Zn(II) demonstrated the capability of this ion to replace the Cu(I) ion into the fully loaded protein and to restore the folded Zn(II)-Ros87 protein.

In the case of Cu(II), the binding followed via UV-Vis also reaches a plateau at 1:1 metal/protein ratio, although with a binding constant lower with respect to that measured for Cu(I). Also in this case, the NMR data demonstrate the presence of a monomeric protein in solution, while on the contrary, the back-titration of Cu(II)-Ros87 with the Zn(II) ion shows a clear displacement of the Cu(II) ion by Zn(II) that does not restore the structure of Zn(II)-Ros87. The overall behavior indicates that the Cu(II) ion triggers the oxidation of the cysteines and the formation of an intramolecular disulfide bridge. This is in accordance with HSAB: since histidine is a borderline ligand, it has the capacity to bind both Cu(I) and Cu(II), while the soft cysteine ligand binds Cu(I) more effectively. Our data, interpreted in light of the above-reported literature findings, indicate that copper’s affinity for small zinc-binding domains is generally governed by metal-to-ligand interaction. As a result, in a small zinc finger peptide, Cu(I) is able to displace the native zinc ion, demonstrating that the magnitude of the binding constant is dominated by the metal-to-ligand interaction that prevails on any other stabilization effect arising from the other folding driving forces. The functional and structural fate of the substituted domain relies upon the well-known structural diversity of the zinc finger domain. Metal ion affinities and domain fold data in these cases, though, only partially reflect the binding properties of a larger protein construct. In Ros87, we have demonstrated that the structuring effect of the metal-binding event is not sufficient to lead to concomitant protein folding in the correct functional domain; this is clearly reflected by the magnitude of the determined binding affinities strongly influenced by Ros87’s larger hydrophobic core. In the presence of Cu(I) and Cu(II), the structuring effect of the hydrophobic core is not enough to consent the protein to fold, causing meaningfully higher Kd* values.

We have demonstrated that Ros87 folds using a bipartite mechanism that needs the formation of a Zn(II) binding intermediate formed by the two cysteines and the second coordinating histidines [32]. Natural homologs of Ros87 lack the structural Zn(II) ion and, although less stable, achieve the same functional fold, suggesting a minimal cost of protein folding compared to the free-energy contribution of Zn(II) binding. However, this cost, although small, appears to be crucial to determine the final stability and function of Ros87. Ros87’s coordination mode of Cu(I) and Cu(II) evidenced by our data suggests the inability of these two ions to stabilize the proper folding intermediate and, as a result, in the presence of both metals, the protein remains substantially unstructured. Thus, we can conclude that the selection of the proper metal ion depends upon properties that mutually influence one another. Indeed, the identity and the number of metal-interacting residues, their interaction with the secondary coordination sphere, the interplay between residues not directly involved in the coordination sphere and their chemical surroundings could ultimately fine-tune the tertiary functional structure of the protein.

## 4. Materials and Methods

### 4.1. Protein Expression, Purification and Preparation of the Metal Ion–Ros87 Complexes

^15^N labelled or un-labelled proteins employed for the UV-Vis, CD and NMR experiments were over-expressed and purified. Briefly, the plasmid was introduced into an E. coli host strain BL21(DE3) and the transformed bacteria were plated onto an LB-agar plate containing ampicillin (100 µg/mL). The protein was obtained by growing the cells, adding to the bacterial culture 1.0 mM isopropyl-β-D-thiogalactopyranoside (IPTG) when the absorbance at 600 nm was 0.6 OD. The expression lasted 2 h; for ^15^N labeling, the cells were grown at 37° in a minimal medium in which the only nitrogen source was 0.5 g/L ^15^NH_4_Cl Then, the cells were harvested by centrifugation (3750 rpm for 40 min) and the pellet was resuspended in 20 mM Na_2_HPO_4_ (pH 6.8) buffer. The suspension was lysed by sonication and centrifuged at 16,500 rpm for 40 min. The supernatant was filtrated with a 0.22 µm filter membrane to remove cell debris and applied to a Mono S HR 5/5 cation exchange chromatography column (Amersham Bioscences, Amersham, UK) equilibrated with phosphate buffer. The fractions with the proteins were collected and applied to a HiLoad 26/60 Superdex 75 (Amersham Bioscences, Amersham, UK) gel filtration chromatography column equilibrated with 20 mM Na_2_HPO_4_ (pH 6.8), 0.2 M NaCl.

An Amicon ultra-15 (Merck, Burlington, MA, USA) centrifugal filter was used to concentrate the proteins after the purification step to reach the desired final concentration. The zinc ion was removed from the samples by acidifying native Zn(II)-Ros87, dialyzing with 10 mM Tris buffer with different concentrations of TCEP at pH 3 (150µM for Cu(I) and 50 µM for Cu(II)). Then, the pH was fixed at 6.5 and the samples dialyzed with 10mM Tris 150µM TCEP for Cu(I) and 10 mM Tris buffer solution for Cu(II).

### 4.2. UV-Vis Spectroscopy

UV-Vis spectra were recorded at room temperature using a Shimadzu UV-1800 spectrophotometer in the range 200–800 nm; for Cu(I)-Ros87, we used 10 mM Tris, 150 µM TCEP at pH 6.5 buffer, while for Cu(II)-Ros87, we used 10 mM Tris at pH 6.5. Cu(II) catalyzes the oxidation of TCEP in TCEPO reducing to Cu(I); the reaction is immediate [41]. The TCEP remains in abundant excess in the experiments involving Cu(I)-Ros87 in order to reduce the cysteines.

Apo-Ros87 was obtained with dialysis of Zn(II)-Ros87 and its concentration was determined by the absorbance at 280 nm at pH 3 using a molar absorption coefficient of 9970 M^−1^ cm^−1^ (www.expasy.org, accessed on 16 October 2021). 

In the case of Cu(I)-Ros87, to obtain a binding curve and to estimate the Cu(I) binding affinity, we acquired a direct titration starting from Apo-Ros87 (~10 µM) with CuCl_2_ solution (5 mM) up to 2.4 Cu(I):protein ratio. We followed the absorption band at 263nm to estimate the dissociation constant of Cu(I)-Ros87 complex [42]. Similarly, in the case of Cu(II)-Ros87, we acquired a direct titration starting from Apo-Ros87 (~10 µM) with CuCl_2_ solution (5 mM) up to 2.4 Cu(I):protein ratio. Here, to estimate the dissociation constant of Cu(II)-Ros87 complex, we followed the absorption band at 236 nm. All of the data obtained were fitted using the following binding isotherm [54]:$$f_{pCu}=\frac{A_{x}-A_{0}}{A_{max\;}-A_{0}}=\frac{(K_{d}^{Cu}+\;{\left[P\right]}_{tot}+{\left[Cu\right]}_{tot})-\sqrt{{\left({\left[P\right]}_{tot}+{\left[Cu\right]}_{tot}+K_{d}^{Cu}\right)}^{2}-4{\left[P\right]}_{tot}{\left[Cu\right]}_{tot}}}{2{\left[P\right]}_{tot}}$$
where *f_pCu_* is the fractional saturation; *A_0_* and *A_max_* are the absorbance values in the absence and presence of copper, respectively; [*Cu*]*_tot_* is the total concentration of the Copper(I) or Copper(II) added; *K_d_^Cu^* is the apparent dissociation constant of the Copper(I)/Copper(II)-protein complex, respectively; and [*P*]*_tot_* is the protein concentration.

Furthermore, to evaluate how zinc displaces copper, a reverse titration of Cu(I)-Ros87 and Cu(II)-Ros87 with ZnCl_2_ (5 mM) was performed up to a Zn(II):Me-Ros87 ratio of 2.4:1. The titration of Copper-Ros87 with Zn induces a decrease in the characteristic absorption band of the two xenobiotic metals. To estimate the dissociation constant of the Copper-Ros87 complex, the fractional saturation values were calculated. The data obtained were fitted using the following equation [55]:$$f_{pZn}=\frac{A_{x}-A_{max}}{A_{0}-A_{max}}=\frac{(K_{d}^{Zn}{\left[P\right]}_{tot}+K_{d}^{Zn}{\left[Cu\right]}_{tot}+K_{d}^{Cu}{\left[Zn\right]}_{tot}-K_{d}^{Cu}{\left[P\right]}_{tot})-\sqrt{{\left(K_{d}^{Zn}{\left[P\right]}_{tot}+K_{d}^{Zn}{\left[Cu\right]}_{tot}+K_{d}^{Cu}{\left[Zn\right]}_{tot}-K_{d}^{Cu}{\left[P\right]}_{tot}\right)}^{2}-4{\left[P\right]}_{tot}\left(K_{d}^{Zn}-K_{d}^{Cu}\right)K_{d}^{Zn}{\left[Cu\right]}_{tot}}}{2{\left[P\right]}_{tot}\left(K_{d}^{Zn}-K_{d}^{Cu}\right)}$$
where *f_pZn_* is the fractional saturation; *A*_0_ and *A_max_* are the absorbance values in the absence and presence of Zinc(II), respectively; [*Zn*]*_tot_* is the total zinc concentration; [*Cu*]*_tot_* is the total concentration of the Copper(I) or Copper(II) added; *K_d_^Zn^* and *K_d_^Cu^* are the apparent dissociation constants of the Zn(II)–protein complex and Copper(I)/Copper(II)–protein complex, respectively. Titration was performed at the lowest concentration possible to obtain a binding isotherm and to avoid metal binding at full saturation. The fitting of the data gave a good R-square in all cases. The data were fitted using the program GraphPad Prism 7.0.

### 4.3. CD Spectroscopy

Circular dichroism (CD) spectra of the Ros87 protein complexed to the different metals were acquired on a JASCO J-815 CD spectropolarimeter equipped with a Peltier temperature control. The data were collected using a quartz cuvette with a 1 cm path-length in the 200–260 nm wavelength range with a data pitch of 1 nm, a band width of 1 nm and scanning speed of 50 nm/min.

Protein samples were prepared in 10 mM Tris, 150 μM TCEP at pH 6.5 for Cu(I)-Ros87 and with 10 mM Tris at pH 6.5 for Cu(II)-Ros87. The correct reduction of Cu(II) to Cu(I) was evaluated via UV-Vis before starting the CD experiment. The experiments were conducted on ~8/µM of Ros87 samples and fresh solutions of 5.0 mM CuCl_2_ were used to fully load the apo-proteins up to a final copper:protein ratio of 2.4:1.

### 4.4. NMR Spectroscopy

The NMR spectra of the fully loaded proteins, in 10 mM Tris, 150 µM TCEP, at pH 6.5, were recorded at 298 K on a Bruker Advance III HD 600 MHz equipped with a triple-resonance Prodigy N2 cryoprobe with z-axis pulse field gradient.

The NMR samples contained 250 µM (into 250 μL of 90% H_2_O/10% ^2^H_2_O) of ^15^N-labelled Ros87 protein; zinc and copper fully loaded NMR samples were obtained after the addition of a 1:1 ratio of CuCl_2_/Ros87. The buffer for the NMR experiments was 10 mM Tris 1mM TCEP pH 6.5 to obtain a total reduction of copper in solution.

For both metals, the tautomeric state of histidine residues was identified by acquiring ^1^H-^15^N-HSQC experiments using an INEPT transfer delay tm = (1/4J) = 11.4 ms to obtain the coherence transfer from ε1H and δ2H protons to the ε2N and δ1N through the ^2^J scalar coupling constant. The spectral width (SW) values were 6002.201 Hz and carrier at 4.69 ppm (*t*_2_) and 6690.341 Hz and carrier at 165 ppm (*t*_1_).

The experimental matrix was 1024 × 90 complex data points and was transformed to yield a final matrix of 2048 × 1024 data points.

The spectra were processed with the Bruker topspin software and ^1^H, and ^15^N chemical shifts were calibrated indirectly by using external reference.

The relaxation rate R_2_ was estimated by analyzing in the [^1^H-^15^N] HSQC spectrum the ^15^N linewidth (λ), that is, the full peak width at half maximum height. λ is directly related to the transverse relaxation rate R^2^ (s^−1^) = λ/2 (Hz) and it reports on the dynamic proprieties of the molecules in solution.

The PDB structures were visualized and analyzed using the software Chimera [40].

### 4.5. Chemical Shift Evaluation

The chemical shift analysis for the copper-complexed Ros87 as well as for the Apo-Ros87 was performed by evaluating the correlation between the ^1^H_N_ and ^15^N shifts for each sample. In particular, here, the ^15^N chemical shifts were weighted by multiplying the ^15^N shifts by a scaling factor α, which corresponds to the ratio between the range of backbone ^1^HN and the range of ^15^N chemical shifts:α=highest H1 C.S. ppm−lowest H1 C.S. (ppm)(highest N15 C.S. ppm−lowest N15 C.S. (ppm)

Only diagnostic signals arising from backbone residues were included in the analysis.

### 4.6. EMSA (Electrophoretic Mobility Shift Assay)

The EMSA experiments were carried out as previously reported [56]. Briefly, 25 pmol of Apo-Ros87 and fully loaded metals-Ros87 was incubated for 10 min on ice with 2.5 pmol of double-stranded oligonucleotide VirC in binding buffer (25 mM HEPES pH 7.9, 50 mM KCl, 6.25 mM MgCl_2_, 5% glycerol). The samples were loaded onto 5% polyacrylamide gel, and electrophoresis was performed at room temperature for 75 min. The gels were then stained with diamond nucleic acid dye (Promega) and imaged with a Typhoon Trio+ scanner (GE Healthcare).

## Figures and Tables

**Figure 1 ijms-23-11010-f001:**
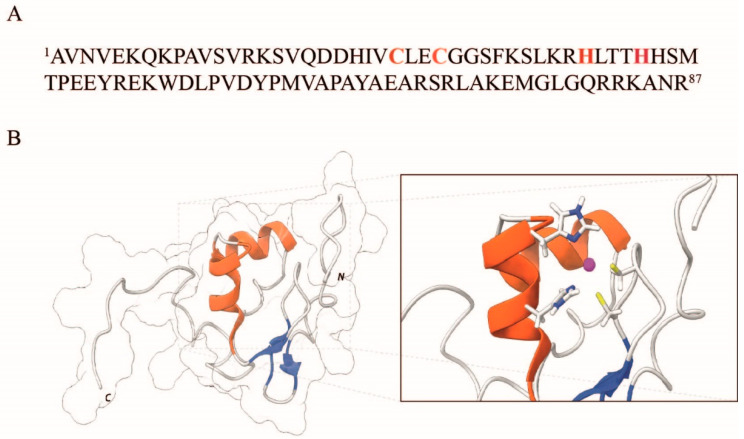
(**A**) Amino acid sequence of Ros87. The coordinating residues are highlighted in red. (**B**) NMR representative structure of Ros87 (PDB ID: 2JSP), realized using the software Chimera [40]. The secondary structure elements are depicted in red (α-helix) and blue (β-strands). A close view of the side chains of coordinating residues is shown in the inset.

**Figure 2 ijms-23-11010-f002:**
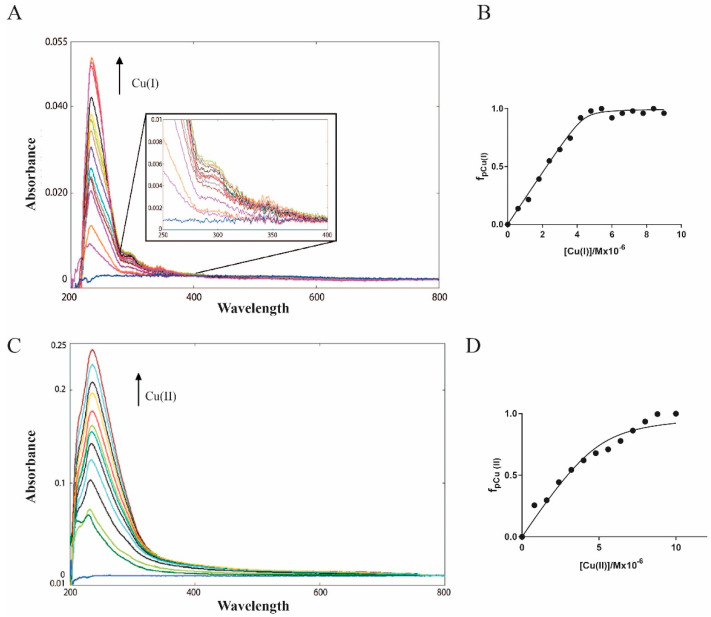
UV-Vis spectra of Apo-Ros87 with (**A**) Cu(I) and (**C**) Cu(II), respectively; fitting curves for (**B**) Cu(I)-Ros87 and (**D**) Cu(II)-Ros87.

**Figure 3 ijms-23-11010-f003:**
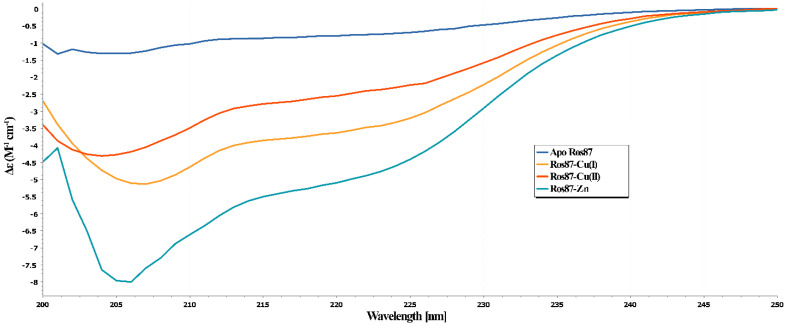
Far-UV CD spectra of Apo-Ros87 (blue line), Zn(II)-Ros87 (yellow), Cu(I)-Ros87 (orange), and Cu(II)-Ros87 (light blue). All of the spectra were acquired at 298K and normalized for concentration.

**Figure 4 ijms-23-11010-f004:**
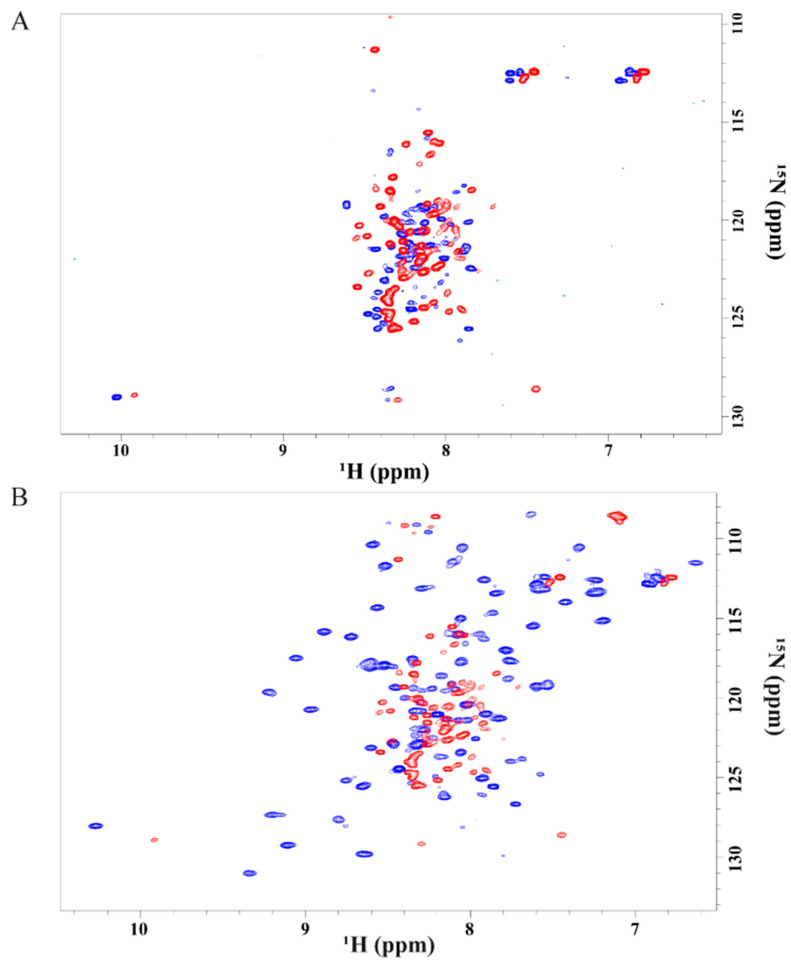
(**A**) Overlay of the ^1^H−^15^N HSQC spectrum acquired for Apo-Ros87 (blue) with Cu(I)-Ros87 (red) at 298K. (**B**) Overlay of the ^1^H−^15^N HSQC spectrum acquired for Zn-Ros87 (blue) with Cu(I)-Ros87 (red) at 298K.

**Figure 5 ijms-23-11010-f005:**
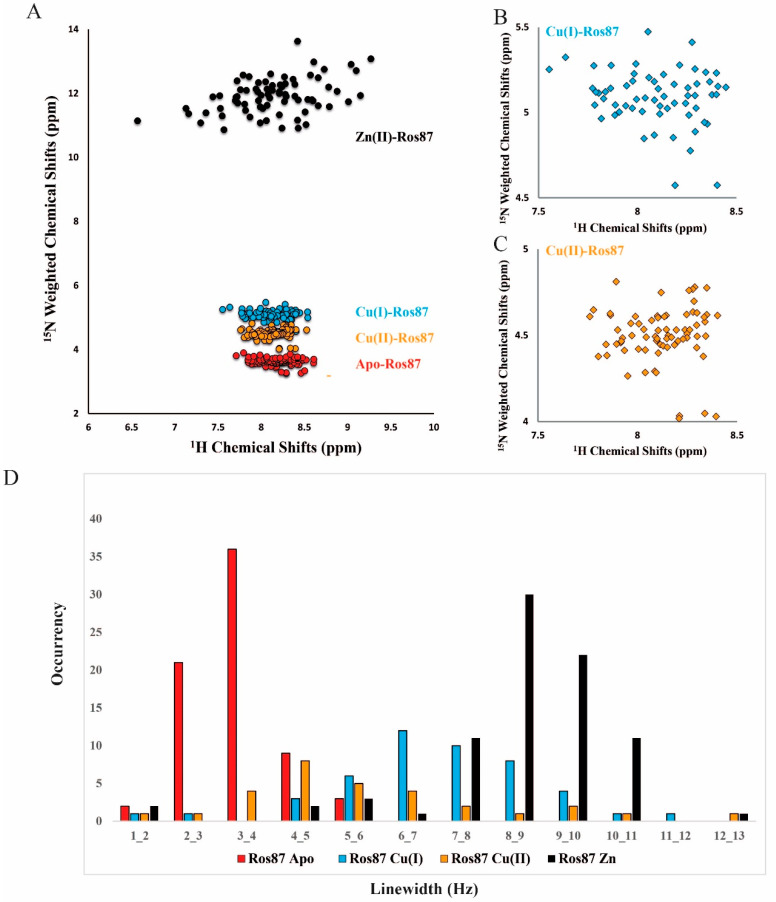
Chemical shift analysis. (**A**) Correlation plots of the ^15^N-weighted and ^1^H chemical shifts for Apo-Ros87 (red), Cu(I)-Ros87 (blue) and Cu(II)-Ros87 (light green). Close view of the ^15^N-weighted/^1^H shifts plots obtained for (**B**) Cu(I)-Ros87 and (**C**) Cu(II)-Ros87. (**D**) Analysis of relaxation rates R2 for Apo-Ros87 (red), Cu(I)-Ros87 (blue), Cu(II)-Ros87 (light green) and Zn-Ros87 (black).

**Figure 6 ijms-23-11010-f006:**
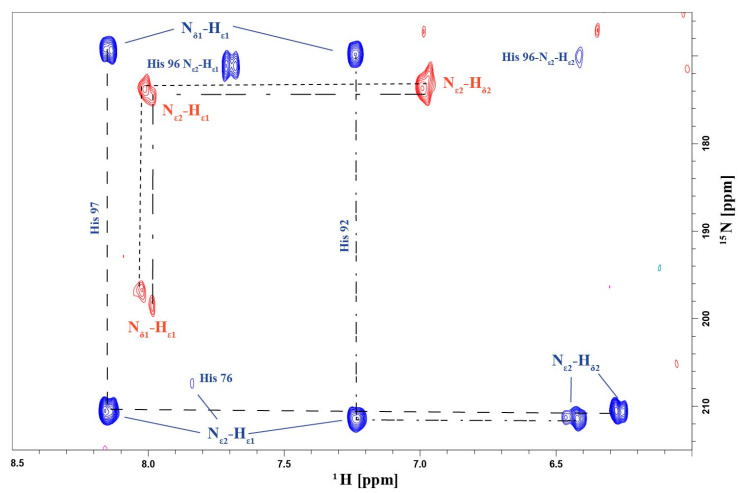
The ^1^H-^15^N-HSQC spectrum in which only HN/N two-bond correlations are reported for Cu(I) Ros-87 (red) and Zn-Ros87 (blue).

**Figure 7 ijms-23-11010-f007:**
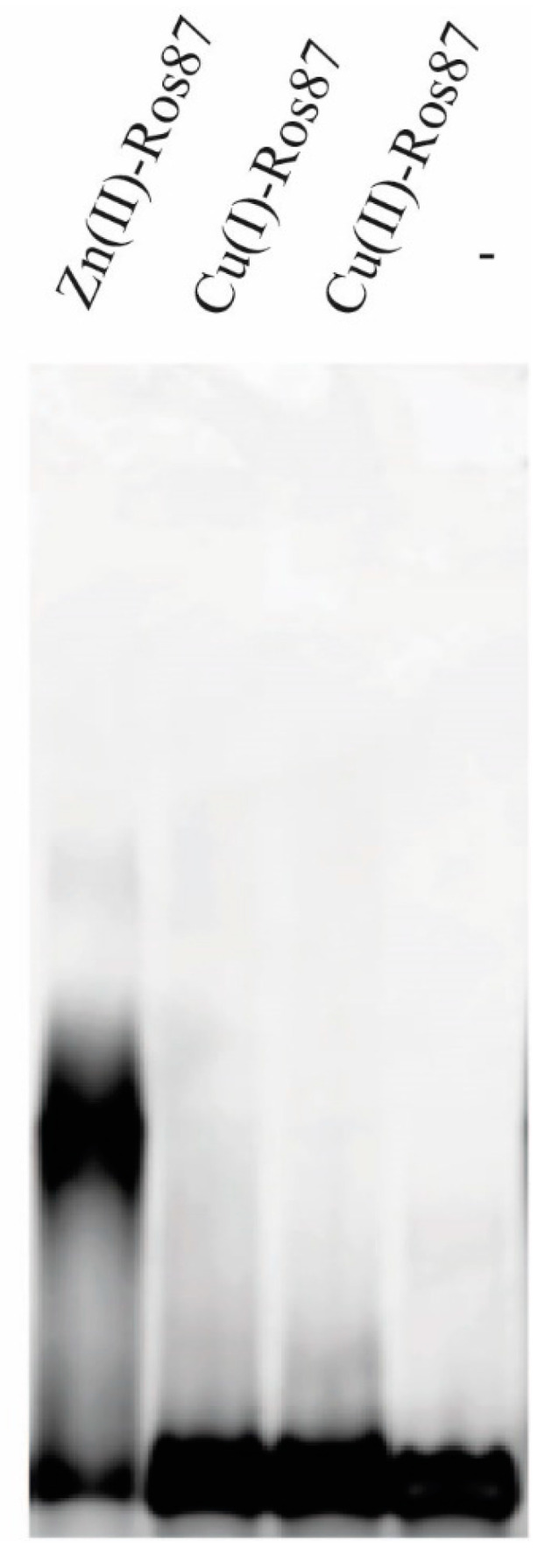
EMSA experiments. Each protein tested is indicated on the top of the image.

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
