# Peer review of "Copper (I) or (II) Replacement of the Structural Zinc Ion in the Prokaryotic Zinc Finger Ros Does Not Result in a Functional Domain"

_ijms, 2022, doi:10.3390/ijms231911010_

Round 1

Reviewer 1 Report

Review to Dragone et al, Copper (I) or (II) replacement of the structural zinc ion, IJMS

The paper by Dragone et al. uses spectroscopic technique to study copper binding to a Zinc finger protein. The data reveal that the protein remains unfolded after addition of Cu(I) and Cu(II) ions despite high affinity of these metal ions for the protein. The topic is certainly of general interest.

Unfortunately this seems to be a very rapidly assembled paper that contains many errors. I also feel that the novelty is somewhat limited, given the much better written and much more detailed paper by the same group reporting on Ni, Mg and Pb ions binding to the same protein. In addition the same topic, although without suing NMR, was already investigated in reference 41. For that reason I am opting for rejection, or a complete resubmission in which errors are removed but that also adds substantially more data/experiments.

Specific points:

The titrations followed by UV to derive Kds for metal binding: The first equation 4 is certainly wrong. The term in front of the root is a sum and does not contain any products (such as Kd*[M]tot). The closing bracket is missing. The second equation on the same page is unclear to me. It is referenced to paper 42, but nowhere in that paper this equation is shown. For similar Kd’s of Cu and Zn this equation would go to infinity (denominator becomes zero), so there must be something wrong there as well. How can you reliably fit against wrong equations?

The UV data show strong binding. However, the curve in Fig. 2C is certainly done in the titration regime (concentration of protein should be around the Kd, but the data show that it must be much higher, this is not a proper binding curve from which a Kd can be derived (see the nice review of Hulme , British Journal of Pharmacology (2010), 161, 1219–1237).  Also the data in Fig. 2D look a bit strange like a 2-step procedure.

My biggest problem is believing that a protein that seems to bind Cu ions so well is entirely unstructured. Also, the CD data indicate structure, and that should be seen in the proton chemical shifts. I wonder whether something happened during preparation of the NMR sample? Was a UV spectrum recorded on the NMR sample? I also think that there are better NMR methods available to determine residual structure (15N{1H}-NOE, RDCs, PCS, even T2 rates), and this was done much better in ref. 42 by the same authors.

No units were given for Kds throughout the paper

I don’t think the introduction of the scaling parameter alpha helps with the analysis. I wonder whether the difference is really significant, or some unspecific salt effect. I also noticed that there is one signal at 7.5/129 ppm in only the Cu(I) species, which may cause a big difference but may not be really of much diagnostic value (likely a signal from the C terminal residue). However, I still see a contradiction between the optical and NMR data, and wonder whether something went wrong with the NMR sample. Or maybe the Cu-complex is unstable at high concentrations or makes something weird there?

Fig. 5D: the binning is difficult to see, this is a poor figure.

A technical point: How were the competition experiments made? Cu(I) is insoluble, so when displacing Zn with Cu this can never be reversible because Cu(I) would precipitate, or?

I believe making mutants would have been interesting, for example Cys->Ser or His->Phe mutants.

I also don’t like figure 1B too much, the original PNAS paper has a much better figure in which the beta sheet is visible. Also, the metal coordination part should be zoomed for better clarity.

Author Response

Reviewer 1

Comment:          Review to Dragone et al, Copper (I) or (II) replacement of the structural zinc ion, IJMS

The paper by Dragone et al. uses spectroscopic technique to study copper binding to a Zinc finger protein. The data reveal that the protein remains unfolded after addition of Cu(I) and Cu(II) ions despite high affinity of these metal ions for the protein. The topic is certainly of general interest.

Unfortunately, this seems to be a very rapidly assembled paper that contains many errors. I also feel that the novelty is somewhat limited, given the much better written and much more detailed paper by the same group reporting on Ni, Mg and Pb ions binding to the same protein. In addition, the same topic, although without suing NMR, was already investigated in reference 41. For that reason, I am opting for rejection, or a complete resubmission in which errors are removed but that also adds substantially more data/experiments.

Answer:

We thank the referee for her/his useful comments. We believe that, as also stated by the reviewer, the topic is of general interest and for this reason we have decided to address all the issues raised. The topic was already investigated in reference 41 in which however substantial differences can be found with our model system, being Ros87 a larger domain, and also in the general findings. As stated by the authors of ref 41 “the susceptibility of ZF domains toward Cu(I) substitution will differ among ZFs as well as other zinc metalloproteins. Finally, it should be noted that the Cu(I)-binding behavior of synthetic peptides corresponding to isolated ZF domains may not directly reflect the binding properties of a larger protein construct”. The aim of our work is to complement and expand our and other’s previous studies.

Comment:          Specific points:

The titrations followed by UV to derive Kds for metal binding: The first equation 4 is certainly wrong. The term in front of the root is a sum and does not contain any products (such as Kd*[M]tot). The closing bracket is missing. The second equation on the same page is unclear to me. It is referenced to paper 42, but nowhere in that paper this equation is shown. For similar Kd’s of Cu and Zn this equation would go to infinity (denominator becomes zero), so there must be something wrong there as well. How can you reliably fit against wrong equations?

Answer:

We agree with the referee. The equations reported in the original submission were wrong; however, it was merely a typo. The proper equations are now reported in the new version of the article

Comment:          The UV data show strong binding. However, the curve in Fig. 2C is certainly done in the titration regime (concentration of protein should be around the Kd, but the data show that it must be much higher, this is not a proper binding curve from which a Kd can be derived (see the nice review of Hulme , British Journal of Pharmacology (2010), 161, 1219–1237).  Also the data in Fig. 2D look a bit strange like a 2-step procedure.

Answer:

The UV data have been repeated multiple times, obtaining always the same results. The characterization has been performed at the lowest concentration possible (3M), below which it is not possible to perform the experiments. Data reported in figure 2D have been fitted using different binding models; no other fitting models appropriately described the data.

Comment:          My biggest problem is believing that a protein that seems to bind Cu ions so well is entirely unstructured. Also, the CD data indicate structure, and that should be seen in the proton chemical shifts. I wonder whether something happened during preparation of the NMR sample? Was a UV spectrum recorded on the NMR sample? I also think that there are better NMR methods available to determine residual structure (15N{1H}-NOE, RDCs, PCS, even T2 rates), and this was done much better in ref. 42 by the same authors.

Answer:

We agree with the referee. The protein is not entirely unstructured and both the CD and NMR data nicely reconcile in demonstrating that upon copper binding minor structural rearrangements of the protein can be observed. The NMR spectra for both metal substituted proteins clearly show important chemical shift differences in the proton chemical shift and we have added a sentence in the article to underline this aspect. However, these rearrangements are not sufficient to obtain a fully structured protein. We have added a few sentences in the paper to better specify this finding. Moreover, we have improved the layout of figure 3 as we have realized that the original figure was a bit misleading due to a formatting problem.

It was not possible to record UV spectra on the NMR sample due to the high concentration of the NMR sample. However, we would like to underline that the UV and NMR sample have been prepared with the same procedure and multiple times. The UV and NMR spectra acquired on all the prepared samples have given the same results. So we can confidently exclude accident during the preparation of the samples.

We agree with the referee that better methods are available to determine the residual structure of proteins. In ref 41, the Ni(II) binding by our protein resulted in a well-structured protein with tertiary interactions. We have thus exploited some of these methods to outline the structural differences between the structure of the Ni(II) bound protein and the Zn(II) bound wild type protein. In presence of copper, instead, NMR data clearly show the absence of tertiary interactions. The residual structure of the two proteins could, in principle, still be determined but we believe that such unduly complex work would not change the general message of the article.

 Comment:         No units were given for Kds throughout the paper

Answer:

We have addressed this issue throughout the paper.

Comment:          I don’t think the introduction of the scaling parameter alpha helps with the analysis. I wonder whether the difference is really significant, or some unspecific salt effect. I also noticed that there is one signal at 7.5/129 ppm in only the Cu(I) species, which may cause a big difference but may not be really of much diagnostic value (likely a signal from the C terminal residue). However, I still see a contradiction between the optical and NMR data, and wonder whether something went wrong with the NMR sample. Or maybe the Cu-complex is unstable at high concentrations or makes something weird there?

Answer:

The alpha scaling factor analysis was introduced in the article to show how the NMR data nicely reconcile with the CD data. The protein in presence of both metals, while remaining largely unstructured, undergoes a certain degree of structuration. The effect is different in the two cases and this is nicely shown by the figure. The signal at 7.25/129 was not included in the analysis and this has been specified in the methods section.

Comment:          Fig. 5D: the binning is difficult to see; this is a poor figure.

Answer:

We have modified figure 5D.

Comment:          A technical point: How were the competition experiments made? Cu(I) is insoluble, so when displacing Zn with Cu this can never be reversible because Cu(I) would precipitate, or?

Answer:

We have not displaced zinc with copper; copper was added to the apo-protein. In the competition experiments we only displace the copper from the protein by adding zinc. We demonstrate that the displacement of Cu(I) gives a Zn(II) bound protein that is similar to the protein freshly expressed while the protein obtained is not properly structured when displacing Cu(II).

Comment:          I believe making mutants would have been interesting, for example Cys->Ser or His->Phe mutants.

Answer:

The mutants suggested by the reviewer are certainly interesting. We have studied the effects of the metal binding amino acids substitution in previous studies (PNAS and BBA). We have shown that the Cys->Ser or His->Phe substitution can result in metal lacking proteins. For this reason, we have decided not to mutate Ros87 metal binding site to study the effects of copper binding.

Comment:          I also don’t like figure 1B too much, the original PNAS paper has a much better figure in which the beta sheet is visible. Also, the metal coordination part should be zoomed for better clarity.

Answer:

We have modified figure 1B according to the reviewer suggestions.

Reviewer 2 Report

The study entitled “Copper (I) or (II) replacement of the structural zinc ion in the prokaryotic zinc finger Ros does not result in a functional domain” is not well-presented and well-discussed. It is not obvious what the main goal of this study. Besides, the outcome of this work as “high-resolution description of the interaction of copper with Ros demonstrating that copper, in both oxidation states, binds the protein but the replacement does not give rise to a functional domain” is not sufficient to be published in IJMS. There is not any advantage of replacing of zinc with copper.

Here are other points:

·       In these following sentences: “is” must be “are”.

-For this reason, the results of copper interaction with zinc binding proteins is of great interest.

- For instance, copper interferences with the DNA-binding activity of zinc finger proteins is associated with the development of a lot of diseases.

·       It is not clear Figure 1 belongs to the authors or they downloaded it. If so they must give a reference in Figure 1.

·       There abbreviations in equations were not explained and it is also not clear why authors used these equations and what they found as a value and this value affected which problem.

-Both metal ions have arisen as non-structural intracellular mediator of cell signaling. In this sentence, “mediator” must be “mediators”.

·       Is there any selective mode of action? Authors indicated the tolerance of the Zn to Cd substitution but not the replacement of the wild type metal by Ni(II), Pb(II) and Hg(II). What about the other redundant metals such as Na, K and Ca?.

·       The direct titration of Apo-Ros87 with Cu(II) is reported in Figure 1 (Panel C) together with the plot of absorbance versus concentration of Cu(II) (panel D). In this sentence Figure 1 must be Figure 2.

·       In Discussion, authors mentioned from other studies like: Yuan and co-workers demonstrated the capability of Cu(I) to displace Zn(II) in Sp1, giving a wellfolded Copper-Finger-Protein unable however to bind its target DNA [13]. Another study has shown that Zn(II) displacement by Cu(I) did not affect the structure and DNA binding affinity of the non-classical zinc finger domain of transcription factor CRR1 (the copper response regulator 1)[53]. However, authors did not discuss what the correlation of these studies with this study or advantages or disadvantages.

Author Response

Reviewer 2

Comment:          The study entitled “Copper (I) or (II) replacement of the structural zinc ion in the prokaryotic zinc finger Ros does not result in a functional domain” is not well-presented and well-discussed. It is not obvious what the main goal of this study. Besides, the outcome of this work as “high-resolution description of the interaction of copper with Ros demonstrating that copper, in both oxidation states, binds the protein but the replacement does not give rise to a functional domain” is not sufficient to be published in IJMS. There is not any advantage of replacing of zinc with copper.

Answer:

We thank the referee for having outlined these points. The main goal of our article is to complement our and other’s previous article in the study of the effects of metal ion replacement in proteins in order to contribute to the general discussion regarding the rules behind the choice of the right metal capable to drive the correct protein folding in metallo proteins. According to the Irving-Williams series our model protein should preferentially bind copper. However, we demonstrate that the protein is capable of binding this metal ion in both oxidation states but the binding does not lead to a functional domain. Thus, we demonstrate that the identity and the number of metal-interacting residues are not sufficient to explain the selection by a protein of a metal ion rather than another. We agree with the reviewer that there is not advantage of replacing zinc with copper and as such we try to demonstrate that this lack of advantage resides in the fine tuning of multiple properties in the protein. We believe that the topic of our article is certainly of general interest and worth of publication in IJMS. For these reasons, we have tried to better outline these concepts throughout the article modifying it in the discussion and introduction sections. We believe that the referee will find this new version of our paper better presented and discussed.

Comment:             Here are other points:

   In these following sentences: “is” must be “are”.

-For this reason, the results of copper interaction with zinc binding proteins is of great interest.

- For instance, copper interferences with the DNA-binding activity of zinc finger proteins is associated with the development of a lot of diseases.

Answer:

We have addresses the issues raised.

Comment:          It is not clear Figure 1 belongs to the authors or they downloaded it. If so they must give a reference in Figure 1.

Answer:

Figure 1 was created using Chimera. We have specified this in figure 1 caption.

Comment:          There abbreviations in equations were not explained and it is also not clear why authors used these equations and what they found as a value and this value affected which problem.

Answer:

We have explained all the abbreviations and commented the results.

Comment:          -Both metal ions have arisen as non-structural intracellular mediator of cell signaling. In this sentence, “mediator” must be “mediators”.

Answer:

We have addresses this issue.

Comment:          Is there any selective mode of action? Authors indicated the tolerance of the Zn to Cd substitution but not the replacement of the wild type metal by Ni(II), Pb(II) and Hg(II). What about the other redundant metals such as Na, K and Ca?.

Answer:

Different metals have different fates when interacting with metal-binding proteins. The quest for the rules of the selection mode of action is still an open question in the study of metal binding proteins. We were not expecting that our model protein was not able to properly fold around copper for what we have reported above. For this reason, we believe that our findings will certainly contribute to the general discussion regarding this topic. It is hard to make previsions about the other redundant metals and we will investigate this topic in the future.

Comment:             The direct titration of Apo-Ros87 with Cu(II) is reported in Figure 1 (Panel C) together with the plot of absorbance versus concentration of Cu(II) (panel D). In this sentence Figure 1 must be Figure 2.

Answer:

We have adjusted this issue in the new version of our manuscript

Comment:          In Discussion, authors mentioned from other studies like: Yuan and co-workers demonstrated the capability of Cu(I) to displace Zn(II) in Sp1, giving a well folded Copper-Finger-Protein unable however to bind its target DNA [13]. Another study has shown that Zn(II) displacement by Cu(I) did not affect the structure and DNA binding affinity of the non-classical zinc finger domain of transcription factor CRR1 (the copper response regulator 1)[53]. However, authors did not discuss what the correlation of these studies with this study or advantages or disadvantages.

Answer:

The susceptibility of zinc finger domains toward Cu(I) substitution differs among ZFs as well as other zinc metallo-proteins. Finally, it should be noted that the Cu(I)-binding behavior of synthetic peptides corresponding to isolated ZF domains may not directly reflect the binding properties of a larger protein construct. Thus, substantial differences can be found between the reported studies and our model system, being the naturally occurring Ros87 a larger domain. As stated above, we have tried to outline differences and analogies in this new version of our manuscript.

Reviewer 3 Report

The paper “Copper (I) or (II) replacement of the structural zinc ion in the prokaryotic zinc finger Ros does not result in a functional domain” by Martina Dragone et al.  characterizes the binding of Cu(I) and Cu(II) to the small protein Ros from A. tumefaciens. The natural ligand of Ros is Zn(II), and the paper shows that its replacement with copper, both Cu(I) or Cu(II), destabilizes the correct folding of the protein, giving rise to a non-functional protein. The authors also suggest that this could be a mechanism of selection of the right cationic species. 

The experiments appear properly designed and the results convincing, the paper is well presented and the results are of medium interest.

Author Response

Reviewer 3

Comment:          The paper “Copper (I) or (II) replacement of the structural zinc ion in the prokaryotic zinc finger Ros does not result in a functional domain” by Martina Dragone et al.  characterizes the binding of Cu(I) and Cu(II) to the small protein Ros from A. tumefaciens. The natural ligand of Ros is Zn(II), and the paper shows that its replacement with copper, both Cu(I) or Cu(II), destabilizes the correct folding of the protein, giving rise to a non-functional protein. The authors also suggest that this could be a mechanism of selection of the right cationic species.

The experiments appear properly designed and the results convincing, the paper is well presented and the results are of medium interest.

Answer:

We thank the referee for her/his appreciation of our work.

Round 2

Reviewer 1 Report

The equations in the paper have been corrected now. The structure Figure 1 is much better now. Otherwise the content has not been changed. I still htink it is a very minor advancement in science over the previous papers from the group. Since the authors have made no effort to improve the science in the paper I have not changed my opinion.

Reviewer 2 Report

Accepted.